# Colistin Resistance Mechanisms in Human and Veterinary *Klebsiella pneumoniae* Isolates

**DOI:** 10.3390/antibiotics11111672

**Published:** 2022-11-21

**Authors:** Manuela Tietgen, Lisa Sedlaczek, Paul G. Higgins, Heike Kaspar, Christa Ewers, Stephan Göttig

**Affiliations:** 1Institute of Medical Microbiology and Infection Control, Hospital of the Johann Wolfgang von Goethe University, D-60596 Frankfurt am Main, Germany; 2University Center of Competence for Infection Control of the State of Hesse, D-60596 Frankfurt am Main, Germany; 3Institute for Medical Microbiology, Immunology and Hygiene, Faculty of Medicine and University Hospital Cologne, University of Cologne, D-50935 Cologne, Germany; 4Federal Office of Consumer Protection and Food Safety, D-10117 Berlin, Germany; 5Institute of Hygiene and Infectious Diseases of Animals, Justus Liebig University Giessen, D-35392 Giessen, Germany

**Keywords:** polymyxin, colistin, *Klebsiella pneumoniae*, insertion element, *mgrB*, lipid A, PmrAB, PhoPQ

## Abstract

Colistin (polymyxin E) is increasingly used as a last-resort antibiotic for the treatment of severe infections with multidrug-resistant Gram-negative bacteria. In contrast to human medicine, colistin is also used in veterinary medicine for metaphylaxis. Our objective was to decipher common colistin resistance mechanisms in *Klebsiella pneumoniae* isolates from animals. In total, 276 veterinary *K. pneumoniae* isolates, derived from companion animals or livestock, and 12 isolates from human patients were included for comparison. Six out of 276 veterinary isolates were colistin resistant (2.2%). Human isolates belonging to high-risk clonal lineages (e.g., ST15, ST101, ST258), displayed multidrug-resistant phenotypes and harboured many resistance genes compared to the veterinary isolates. However, the common colistin resistance mechanism in both human and animal *K. pneumoniae* isolates were diverse alterations of MgrB, a critical regulator of lipid A modification. Additionally, deleterious variations of lipopolysaccharide (LPS)-associated proteins (e.g., PmrB P95L, PmrE P89L, LpxB A152T) were identified. Phylogenetic analysis and mutation patterns in genes encoding LPS-associated proteins indicated that colistin resistance mechanisms developed independently in human and animal isolates. Since only very few antibiotics remain to treat infections with MDR bacteria, it is important to further analyse resistance mechanisms and the dissemination within different isolates and sources.

## 1. Introduction

*Klebsiella pneumoniae* is a Gram-negative pathogen that causes a variety of infections, including urinary tract infections, pneumonia, abdominal infections and sepsis in community and healthcare settings. There are numerous reports on the emergence of multidrug-resistant (MDR) *K. pneumoniae* strains, which have acquired multiple antibiotic resistance determinants such as extended-spectrum beta-lactamases (ESBL) or carbapenemases [1]. The treatment of infections caused by MDR pathogens is a great challenge because of the limited choice of active antibiotics and the lack of quality evidence supporting alternative treatment regimens. Human medicine is therefore strongly dependent on last-line antibiotics, such as colistin (polymyxin E), yet, resistance against colistin is increasing [2]. However, *K. pneumoniae* does not only cause infections in humans but also in livestock and companion animals, e.g., cattle, horses, dogs and cats [3]. In contrast to human medicine, colistin (COL) is one of the most intensively applied antibiotics in European veterinary medicine, particularly for the treatment of gastrointestinal infections in pig, cattle and poultry production [4]. Considering the high importance of COL for the preservation of antibiotic treatment options in human medicine, its use in veterinary medicine has been criticized [5]. The European Medicines Agency (EMA) restricted the administration of COL in veterinary medicine to cases with no other effective alternative and requested member states to reduce their consumption rates drastically by 2020 [5].

The outer membrane of Gram-negative bacteria contains lipopolysaccharides (LPS) in the outermost leaflet, which are stabilized by divalent cations, e.g., Mg^2+^ and Ca^2+^, via electrostatic bridges between lipid A molecules. Colistin is a polycationic lipopeptide antibiotic with amphipathic properties. The primary interaction of COL with the outer membrane is the displacement of Mg^2+^ and Ca^2+^ leading membrane destabilization. Thereafter, two mode of action models have been hypothesized (i) the “self-directed uptake“ and (ii) the “vesicle–vesicle contact pathway” [6,7,8,9]. The first model assumes that the interaction of the positively charged peptide ring of COL with LPS allows the insertion of a second COL molecule in the outer membrane via the N-terminal fatty acyl chain [6,7,8]. Subsequently, the interaction of the lipophilic domains of COL with the fatty acid residues further destabilizes membrane integrity. The lipid tail of COL is also necessary for the disruption of the cytoplasmatic membrane targeting the LPS, which is transported to the outer membrane. The second model hypothesizes that COL might be present in the inner leaflet of the outer membrane [9,10]. The binding of COL to the anionic phospholipids of the cytoplasmic membrane results in an exchange of phospholipids between the membranes, which finally leads to lysis, and bacterial cell death.

Resistance to COL is mediated by inhibition of the interaction between the antibiotic and the lipid A moiety, which is due to either complete loss of LPS or covalent modifications of lipid A so that it is less negatively charged. These lipid A modifications include addition of 4-amino-4-deoxy-L-arabinose (Ara4N) or phosphoethanolamine (PEtN) and are regulated by the two-component regulatory systems PhoPQ and PmrAB [11,12,13]. Upon activation of PhoPQ, the small transmembrane protein MgrB is produced which acts as a negative regulator of this system by feedback inhibition. Chromosomal mutations in the genes encoding this regulator has been shown to cause COL resistance [14]. In 2015, the first plasmid-encoded COL resistance determinant *mcr-1* was discovered and detected in isolates from animals as well as from humans [15,16]. The phosphoethanolamine transferase Mcr-1 catalyses the transfer of PEtN to lipid A which inhibits binding of COL. So far, more than 100 mobilizable *mcr* genes have been identified in different Gram-negative bacterial species [17].

Although COL resistance in *K. pneumoniae* has been increasingly reported for human patients over the last years, COL resistance rates appear to remain low in *K. pneumoniae* isolated from animals [18,19]. Very few studies have analysed the underlying resistance mechanisms of *mcr*-negative *K. pneumoniae* COL-resistant veterinary isolates [17,20,21,22,23,24]. Therefore, the objective of our study was to identify colistin resistance in veterinary isolates, determine the molecular mechanisms and to compare those with isolates obtained from human patients.

## 2. Results

### 2.1. Identification of Colistin Resistance in K. pneumoniae Veterinary Isolates

The objective of this study was to compare COL resistance mechanisms in *K. pneumoniae* isolates of animal and human origin. For this purpose, a total of 276 *Klebsiella pneumoniae* (KP) isolates were collected from veterinary patients in Germany and from dairy cattle and companion animals between 2007 and 2017. Of these 276 isolates, six COL resistant *K. pneumoniae* with MICs of each 32–64 mg/L were identified (6/276; 2.2%): one porcine isolate (IHIT27665), three bovine isolates (IHIT27662, IHIT27663 and IHIT27664) and two canine isolates (IHIT32358 and IHIT33535) (Table 1).

The three bovine isolates were recovered from milk samples of dairy cattle, which suffered from mastitis and were sampled from different farms in Southern and Central Germany. These isolates were resistant to colistin and polymyxin B (Table 2) but exhibited no other acquired resistance (Appendix A). In contrast, IHIT27665 (porcine faeces isolate) and IHIT33535 (canine urine isolate) displayed ESBL phenotypes due to the presence of CTX-M (Appendix A). No history of polymyxin treatment was recorded for the veterinary isolates (Table 1), indicating that antibiotic pressure was not the main driver of resistance development.

Twelve COL-resistant *K. pneumoniae* isolates from human patients were used for comparison (Table 1). These isolates were all MDR according to Magiorakos et al. [25] and expressed ESBLs, and in three cases the carbapenemase KPC (Table 2 and Appendix A). In nine out of 12 human isolates, COL resistance most likely developed independently from selective pressures by COL therapy analogous to the veterinary isolates in our study.

Colistin MICs were determined for all isolates using broth microdilution as the recommended method and additionally with antibiotic gradient tests and the VITEK^®^ 2 system (Table 2 and Appendix A). Notably, all MICs measured with gradient strips and Mueller–Hinton agar were significantly lower compared to the other methods. For example, veterinary isolate IHIT33535 and the human isolates KP_2442, KP_3405 and KP_3996 recorded COL MICs using ETEST^®^ strips with Mueller–Hinton agar of 2 mg/L and hence would have been misinterpreted as susceptible (Appendix A).

In contrast, use of ETEST^®^ strips with Mueller–Hinton E agar, which is a modified Mueller–Hinton agar recommended for ETEST^®^ and non-fastidious organisms, correctly detected COL resistance in all isolates. VITEK^®^ 2 susceptibility testing also correctly revealed that all isolates were COL resistant, but the upper detection limit was 16 mg/L. Colistin MICs as determined by broth microdilution were slightly higher for the human isolates (median: 64 mg/L; range: 8–128) compared to the veterinary isolates (median: 48 mg/L; range: 32–64) but without statistical significance (*p* = 0.49) (Table 2).

### 2.2. Molecular Mechanisms of COL Resistance in K. pneumoniae Isolates

In order to decipher the underlying molecular mechanisms of COL resistance in the veterinary and human isolates, proteins associated with COL resistance or synthesis and modification of LPS were analysed. In 16/18 isolates alterations in MgrB were detected (Table 3). Whereas MgrB point mutations were found in 4/6 veterinary isolates, no point mutations were present in human isolates.

In the veterinary isolates, MgrB point mutations at the nucleotide level of C96T (IHIT33535) or C88T (IHIT27665 and IHIT35358) resulted in premature stop codons. Point mutation A4G at the nucleotide level in IHIT27663 led to the deleterious amino acid change K2E according to the protein analysing software tool PROVEAN [26]. Only a few additional mutations were found which might have had an impact on polymyxin susceptibility (Table 3). For example, isolate IHIT27665 carried the variation T112P in PmrB, which is located in the HAMP domain (present in histidine kinases, adenylate cyclases, methyl accepting proteins and phosphatases), which is also deleterious according to PROVEAN. Hence, both mutations in MgrB and PmrB could lead to COL resistance in this isolate. No deleterious amino acid changes were found in LpxA, LpxL, LpxM or PhoP in any of the isolates.

In human isolates, insertion of different IS elements and large deletions (3481 and 10,934 bp) in MgrB were detected in 9/12 isolates. In KP_3996, MgrB was disrupted at position six by the gene kdgR, which encodes a transcriptional regulator of carbohydrate acid metabolism genes. In the two isolates with an intact MgrB gene, the transferable COL resistance gene *mcr-1* was detected in KP_2442. In KP_03, mutations in PmrB (P95L) and PmrH (P264S) were detected (Table 3). To assess the impact of these two mutations on COL susceptibility, protein structures were predicted using AlphaFold2 and subsequently compared with those of the COL-susceptible reference strain *K. pneumoniae* MGH78578. The superposition of PmrB of *K. pneumoniae* MGH78578 and KP_03 indicated a shift of two alpha helices (R14-E51 and Q56-E104) including the loop in between (Figure 1). The distance between A49 of *K. pneumoniae* MGH78578 and KP_03 was about 8.612 Å and between A47 9.116 Å (Figure 1). This might lead to an altered activity of PmrB, which results in the addition of Ara4N to lipid A. In contrast, the structure of PmrH did not appear to be altered by the P264S amino acid substitution and therefore an impact on COL resistance is rather unlikely (Appendix A).

### 2.3. Genetic Relatedness of COL-Resistant K. pneumoniae from Human and Animal Origin

Multilocus sequence typing (MLST) using the Pasteur scheme revealed five STs (ST15, ST307, ST976, ST1864 and ST163) among the veterinary isolates (Table 1). IHIT27665 and IHIT32358 belong to the high-risk lineage ST15 and IHIT33535 to ST307, both of which have been linked to ESBL and carbapenemase production before [27,28]. The other three isolates belonged to STs that occur only sporadically [29,30,31]. Among the isolates from human patients, ten STs (ST16, ST11, ST48, ST2299, ST437, ST395, ST258, ST15, ST1583 and ST101) were identified (Table 1). All human isolates, except KP_1200 (ST1583) belonged to high-risk lineages (e.g., ST11, ST258 and ST437) or were single locus variants of these (Figure 2). The only common ST between the veterinary and human isolates was ST15. Isolates belonging to the same ST had ≥35 SNPs, but we could not determine any epidemiological links, suggesting that these isolates were not clonal. All other isolates occurred independently from each other, as indicated by the high number of SNPs.

All human isolates displayed an MDR or even XDR phenotype with resistance to 3rd generation cephalosporins and resistance rates of >80% for aminoglycosides, fluoroquinolones and cotrimoxazole in comparison to the veterinary isolates (Table 2 and Appendix A). Likewise, the genome sequences of the animal isolates harboured on average significantly fewer antibiotic resistance genes (median: 4, range: 1–10) compared to the human isolates (median: 13, range: 5–18) (*p* < 0.0042) as determined by the CGE Resfinder tool (Appendix A). Carbapenem resistance in the human isolates was linked to the acquisition of bla_KPC-2_ in KP_1377 and KP_1954 (both ST258) and bla_KPC-3_ in KP_3996 (ST101). Further, the ESBL-encoding gene bla_CTX-M-15_, was detected in 7/12 human isolates (Appendix A).

## 3. Discussion

The objective of this study was to elucidate and compare colistin resistance mechanisms between clinical isolates from animals and humans. Six out of 276 veterinary isolates were COL resistant and displayed MICs of 32 or 64 mg/L (Table 2). Three of these six isolates belonged to high-risk clonal lineages ST15 and ST307 (Table 1), and harboured different acquired resistance genes such as *bla*_TEM-1B_, *tet(A)* or *dfrA* (Appendix A). The two canine isolates IHIT27665 and IHIT33535 carried *bla*_CTX-M_ genes and displayed an ESBL phenotype. The three other isolates were only resistant to ampicillin (intrinsically resistant in *K. pneumoniae*) and polymyxins and likewise did not harbour any known acquired resistance genes. These three isolates belonged to sporadic sequence types and were recovered from dairy cattle with no known prior polymyxin exposure, suggesting that COL resistance occurred before colonisation. In contrast, almost all (11/12) human isolates belonged to high-risk clonal lineages (e.g., ST11, ST258) and were derived from hospitalized patients with severe diseases. These isolates displayed an MDR or XDR (according to Magiorakos et al. [25]) resistance phenotype and harboured significantly more resistance genes, including several ESBL- and KPC-encoding genes, compared to animal isolates.

Although the development of COL resistance in isolates from animals might differ from isolates from humans, the common resistance mechanism seems to be the alteration of the PhoP/PhoQ signalling repressor MgrB. These alterations lead to an addition of Ara4N to lipid A, which results in COL resistance. In 16/18 isolates, *mgrB* was deleted or disrupted by the insertion of IS elements or *kdgR*, whereas point mutations were only found in veterinary isolates (Table 3). Inactivation of *mgrB* due to its disruption by various insertion elements, premature stop codons or deletion of the gene locus have previously been described for several *K. pneumoniae* isolates from humans and one dairy cow [14,20,32]. Inactivation of *mgrB* readily explains COL resistance; however, additional point mutations in several LPS-associated genes (e.g., *pmrAB*, *phoQ*, *lpxB*) were found which might contribute to polymyxin resistance (Table 3).

The plasmid-borne resistance mechanism mediated by *mcr-1* was only detected in one human isolate (KP_2442) [16]. For this mechanism the opportunity of transfer from companion animals or livestock to humans has been described [33,34,35]. However, due to the different mutations in LPS-associated genes as well as the diverse sequence types and high number of SNPs, transmission of COL-resistant strains from animals to humans and vice versa seems to not be the main route of dissemination. In human isolates *mcr-1* is the predominant variant in *E. coli*, whereas in porcine isolates, particularly from Europe and Asia, diverse variants of *mcr-1*–*mcr-9* have been detected [36].

In the human isolate KP_03 neither alteration of MgrB nor acquisition of *mcr-1* was detected. However, the amino acid substitution P95L in PmrB was predicted to result in a non-functional protein. This amino acid is located within the HAMP domain of the protein (amino acids 90–142), which converts conformational changes in the periplasmatic region to the active catalytic cytoplasmatic region of the protein. Molecular modelling revealed a shift of two alpha helices (R14-E51 and Q56-E104) (Figure 1) in this isolate. This might lead to altered activity of PmrB which ultimately results in the addition of Ara4N to lipid A. For example, specific mutations of *pmrB* induce transcription of the *pmrH* (*arnB*) operon, which provokes the addition of Ara4N to lipid A [37]. Likewise, missense mutations in *pmrA* and *pmrB*, particularly in the H-box, lead to an up-regulation of *pmrC*, which is part of the *pmrCAB* operon resulting in the addition of PEtN [38,39]. Further studies are needed to verify if the detected variations in PmrB and the other LPS-associated genes indeed have an effect on COL MIC which is, however, beyond the scope of our study.

In human medicine, COL is used as a last-line antibiotic predominantly in hospitals. In veterinary medicine COL has been regularly used in the past decades, both as therapeutic treatment of Gram-negative gastrointestinal infections in certain food-producing animal species and also for metaphylactic purposes, i.e., for disease prevention. Colistin is typically administered orally and as group treatment. Thus, resistance mechanisms may also develop in healthy animals due to the possible exposure of low COL doses. Under the perspective of the “One Health” concept the use of COL for metaphylaxis should be even more restricted.

## 4. Materials and Methods

### 4.1. Klebsiella Pneumoniae Clinical Isolates

Seventy *K. pneumoniae* isolates recovered from animal specimens were sent to the Institute of Hygiene and Infectious Diseases of Animals at Justus Liebig University in Giessen (Germany) by various veterinary clinics between 2012 and 2017. Additionally, 206 isolates were included, which were part of a monitoring study conducted by the unit of antibiotic resistance monitoring of the Federal Office of Consumer Protection and Food Safety (BVL) in Berlin (Germany) between 2007 and 2013. Overall, these isolates were recovered from dairy cattle (*n* = 201), dogs (*n* = 39), horses (*n* = 23), pigs (*n* = 7) and cat, donkey, ferret, guinea pig, rabbit and sheep (each *n* = 1). Furthermore, twelve COL-resistant isolates, recovered from human patients hospitalized at the Hospital of the Goethe University in Frankfurt am Main (Germany) between 2007 and 2017 were included in the study. Bacterial species identification of isolates was performed using matrix-assisted laser desorption/ionization–time-of-flight (MALDI-TOF) mass spectrometry (bioMérieux, Nürtingen, Germany).

### 4.2. Antimicrobial Susceptibility Testing

Antimicrobial susceptibility was evaluated by determining the minimum inhibitory concentration (MIC) using either antibiotic gradient strips (ETEST^®^, bioMérieux or Liofilchem, Roseto degli Abruzzi, Italy) or the VITEK^®^ 2 system (bioMérieux) employing antimicrobial susceptibility testing cards AST-GN263 and AST-N248. Mueller–Hinton E agar (bioMérieux) was used for colistin gradient strips and Mueller–Hinton agar (Oxoid, Wesel Germany) for all other antibiotics. MICs were interpreted according to EUCAST guidelines V11.0. MIC for COL were determined by broth microdilution as recommended by CLSI and EUCAST (https://www.eucast.org/eucastguidancedocuments/, accessed on 14 November 2022) using cation-adjusted Mueller–Hinton broth (CAMHB) and increasing concentrations (0–256 mg/L) of colistin sulphate salt (Sigma-Aldrich, Taufkirchen, Germany).

### 4.3. Detection and Analysis of Antimicrobial Resistance Determinants

PCR amplification and Sanger sequencing was performed to identify beta-lactamase genes (*bla*_CTX-M_ and *bla*_SHV_), as previously described [40,41]. Whole genome sequences were used to identify other antimicrobial resistance genes using the ResFinder tool version 4.1 (https://cge.food.dtu.dk/services/ResFinder/, accessed on 14 November 2022). For the determination of mutations in genes associated with the synthesis or modification of LPS (*mcr-1*, *mgrB*, *pmrA*, *pmrB*, *phoP* and *phoQ*), PCR and Sanger sequencing with primers listed in Appendix A were employed. Amino acid sequences for KdtA, LpxA, LpxB, LpxH, LpxK, LpxL, LpxM, PmrC, PmrE and PmrH were derived from whole genome sequences. Variations of proteins and alterations in biological function were analysed employing the Protein Variation Effect Analyzer (PROVEAN) platform (http://provean.jcvi.org/index.php, accessed on 14 November 2022) and *K. pneumoniae* MGH78578 (GenBank: CP000647.1) as a reference. Protein variations with a score of −2.5 were considered as “deleterious” [26]. Domains within the proteins were compared to the annotations from the UniProt platform (http://www.uniprot.org/, accessed on 14 November 2022). Structure prediction analysis of PmrB and PmrH was performed employing the software ChimeraX v1.4 and the structure prediction function via Colab/AlphaFold2 as described [42,43].

### 4.4. Whole Genome Sequencing and Analysis of Genetic Relatedness

For whole genome sequencing of the *K. pneumoniae* isolates, DNA was extracted with the Master Pure^TM^ DNA Purification Kit (Biozym Scientific, Hessisch Oldendorf, Germany). Genome sequencing was carried out with an Illumina MiSeq with 300 bp paired end reads with an average coverage of 70–120×. De novo assembly after quality trimming of the reads was conducted using the CLC Genomics Workbench v. 9 (CLC bio, Aarhus, Denmark) with standard parameters, scaffolding and exclusion of contigs smaller than 200 bp. Sequence types were determined according to the Pasteur scheme (http://bigsdb.pasteur.fr/klebsiella/klebsiella.html, accessed on 14 November 2022). Clonality of colistin-resistant *K. pneumoniae* isolates was investigated by core genome MLST (cgMLST) using Ridom SeqSphere+ (Ridom, Münster, Germany) [44].

## Figures and Tables

**Figure 1 antibiotics-11-01672-f001:**
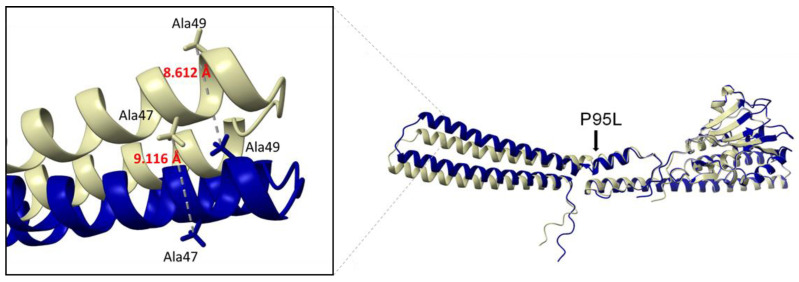
Superposition of PmrB of *K. pneumoniae* MGH78578 in blue (accession number WP_004179093) and PmrB of KP_03 in beige. Amino acid substitution P95L is marked by an arrow.

**Figure 2 antibiotics-11-01672-f002:**
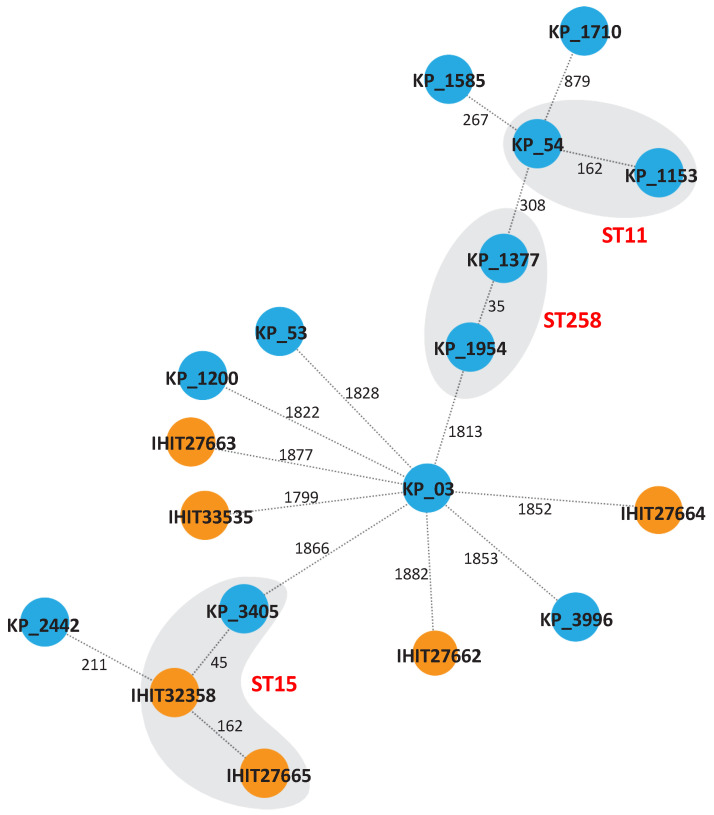
Genetic relatedness of colistin-resistant veterinary and human *K. pneumoniae* isolates. The minimum spanning tree was generated by cgMLST using SeqSphere (Ridom) based on 2358 alleles of the core genome. Numbers between the nodes indicate the number of different alleles. Veterinary isolates are depicted in orange; human isolates are shown in blue. Affiliation to the same sequence is indicated by the grey background.

**Table 1 antibiotics-11-01672-t001:** Epidemiological data of *K. pneumoniae* isolates resistant to COL obtained from veterinary and human patients.

**Strain**	**Year**	**Disease**	**Animal**	**Specimen**	**Colistin** **Treatment**	**Sequence** **Type (ST)**
IHIT27665	2012	Unknown	Swine	Faeces	-	15
IHIT27662	2012	Mastitis	Dairy cattle	Milk	-	976
IHIT27663	2011	Mastitis	Dairy cattle	Milk	-	1864
IHIT27664	2007	Mastitis	Dairy cattle	Milk	-	163
IHIT32358	2016	Otitis	Dog	Ear swab	unknown	15
IHIT33535	2017	Cystitis	Dog	Urine	unknown	307
**Strain**	**Year**	**Disease**	**Detection Site/Specimen**		**Colistin** **Treatment**	**Sequence** **Type (ST)**
KP_03	2011	Heart surgery	BAL		-	48
KP_53	2010	Kidney transplantation	Abdominal swab		Yes	16
KP_54	2007	Liver cirrhosis	Wound swab		-	11
KP_1153	2011	ARDS	Bronchial secretion		-	11
KP_1200	2011	Renal insufficiency	Peritoneal fluid		-	1583
KP_1377	2011	Sarcoma	Urine		-	258
KP_1585	2013	AML	Blood		Yes	437
KP_1710	2013	Ileus	Tracheal secretion		-	395
KP_1954	2014	Pancreatitis	Bronchial secretion		Yes	258
KP_2442	2015	ALL	Blood		-	2299
KP_3405	2016	Liver cirrhosis	Rectal swab		-	15
KP_3996	2017	Cardiomyopathy	Blood		-	101

BAL = bronchoalveolar lavage; AML = acute myeloid leukaemia; ARDS = acute respiratory distress syndrome; ALL = acute lymphatic leukaemia.

**Table 2 antibiotics-11-01672-t002:** Minimum inhibitory concentrations (MIC) for COL-resistant clinical *K. pneumoniae* isolates. MIC were determined employing antibiotic gradient strips unless otherwise indicated. MIC indicating resistance according to EUCAST breakpoints are highlighted in bold.

	MIC (mg/L)																	
	Veterinary Isolates						Human Isolates											
Antibiotic Agent	IHIT27662	IHIT27663	IHIT27664	IHIT27665	IHIT32358	IHIT33535	KP_03	KP_53	KP_54	KP_1153	KP_1200	KP_1377	KP_1585	KP_1710	KP_1954	KP_2442	KP_3405	KP_3996
Piperacillin/tazobactam	2	2	1	**64**	**16**	8	**>256**	**>256**	**>256**	**>256**	**>256**	**>256**	**>256**	**>256**	**>256**	**32**	**>256**	**>256**
Cefotaxime	0.064	0.064	0.064	**64**	0.064	**128**	**>256**	**>256**	**>256**	**>256**	**>256**	**>256**	**>256**	**>256**	**>256**	**128**	**>256**	**>256**
Ceftazidime	0.25	0.25	0.125	**8**	1	**16**	**48**	**>256**	**>256**	**>64**	**>64**	**>256**	**>64**	**64**	**>256**	**32**	**>64**	**>256**
Imipenem	0.032	0.25	0.064	0.25	0.125	0.25	0.25	**>32**	**>32**	2	1	**32**	2	0.25	**>32**	0.125	1	**>32**
Meropenem	0.032	0.032	0.125	1	0.032	0.064	0.064	**>32**	**>32**	**16**	4	**32**	4	0.125	**>32**	0.032	8	**>32**
Tobramycin	1	1	1	**16**	1	**8**	**32**	**64**	**32**	**32**	**32**	**64**	**32**	**32**	**32**	**16**	**8**	**>256**
Amikacin	2	2	1	2	2	2	4	**16**	**16**	**64**	**32**	**64**	8	8	**32**	4	2	**>256**
Ciprofloxacin	0.016	0.032	0.032	**2**	**8**	**16**	**1**	**>32**	**32**	**>32**	**>32**	**>32**	**>32**	**>32**	**>32**	**>32**	**>32**	**>32**
TMP-SMX	0.25	0.25	0.25	2	2	**>16**	**>32**	**>32**	1.5	**>32**	**>32**	**>32**	**>32**	**>32**	**>32**	**>32**	**8**	0.19
Polymyxin B	**4**	**4**	2	**4**	1	1	**8**	2	2	**4**	2	2	2	**4**	**8**	2	1	2
Colistin ^a^	**64**	**32**	**32**	**32**	**64**	**64**	**32**	**64**	**64**	**128**	**32**	**64**	**128**	**128**	**128**	**8**	**32**	**32**

TMP-SMX, Trimethoprim/sulfamethoxazole, ^a^ MIC determined by broth microdilution.

**Table 3 antibiotics-11-01672-t003:** Mutations in putative COL resistance genes in veterinary and human *K. pneumoniae* isolates. Sequence positions (Pos) refer to the COL-susceptible reference strain *K. pneumoniae* MGH78578.

Isolate	MgrB ^a^	IS	PmrA	PmrB ^b^	PmrC	PmrE	PmrH	KdtA	PhoQ	LpxB	LpxH	LpxK	Mcr-1
IHIT27665	Q30Stop	-	-	T112P	-	-	-	-	-	-	-	-	-
IHIT27662	Insertion (Pos +69)	IS903B (1057 bp)	-	-	-	-	-	-	-	-	E234-	L218H	-
IHIT27663	K2E	-	-	-	-	-	-	-	-	-	-	-	-
IHIT27664	Deletion (1253 bp)	-	-	-	-	P89L	-	-	-	-	-	-	-
IHIT32358	Q30Stop	-	G53S	-	-	-	-	-	-	-	-	-	-
IHIT33535	Q33Stop	-	-	-	-	-	-	-	-	-	-	-	-
KP_03	-	-	-	P95L	-	-	P264S	-	-	-	-	-	-
KP_53	Insertion (Pos +60)	IS1R (767 bp)	-	-	-	-	-	-	-	-	-	-	-
KP_54	Deletion (10,934 bp)	-	-	R256G	-	-	-	-	-	-	-	-	-
KP_1153	Insertion (Pos +75)	IS*Ecp*1 (1656 bp)	-	R256G	-	-	-	-	-	-	-	-	-
KP_1200	Deletion (3481 bp)	-	-	-	I83T	-	-	-	-	-	-	-	-
KP_1377	Insertion (Pos +133)	IS*Kpn*25 (8154 bp)	-	R256G	-	-	-	-	-	-	-	-	-
KP_1585	Deletion (6378 bp)	-	-	R256G	-	-	-	-	-	-	-	-	-
KP_1710	Deletion (1300 bp)	-	-	R256G	-	-	-	A180T	-	A152T	-	-	-
KP_1954	Insertion (Pos +74)	IS*Kpn*26 (1195 bp)	-	R256G	-	-	-	-	N253P	-	-	-	-
KP_2442	-	-	-	-	S257L	-	-	-	-	-	-	-	+
KP_3405	Deletion (12,160 bp)	-	-	-	S257L	-	-	-	-	-	-	-	-
KP_3996	Insertion (Pos +6)	*kdgR* (1147 bp)	-	-	-	-	-	-	-	-	-	-	-

IS, insertion element; ^a^ for isolate IHIT27664 the deleted region equated to position 2563834–2565087 in MGH78578, for KP_54 to 2555901–2566835, for KP_1200 to 2561556–2565037, for KP_1585 to 2562404–2568782, for KP_1710 to 2563833–2565133 and for KP_3405 to 2556097–2568259. ^b^ Alphafold structures of PmrB variants T112P and R256G are shown in Appendix A, respectively.

## Data Availability

The genome sequences of all isolates were deposited publicly in NCBI [BioProject ID PRJNA901370].

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
