# Peer review of "Colistin Resistance Mechanisms in Human and Veterinary Klebsiella pneumoniae Isolates"

_antibiotics, 2022, doi:10.3390/antibiotics11111672_

Round 1

Reviewer 1 Report

Manuscript ID: antibiotics-2008465

General and Specific comments:

The study is original, contributing to the existing knowledge regarding colistin resistance mechanisms among Klebsiella pneumoniae from human and animal origin.

Some points that deserve revision are detailed below:

ABSTRACT

1. Lines 17 and 23: Please replace by “multidrug-resistant”. Please check along the manuscript.

INTRODUCTION

2. Lines 55-61: The mechanisms of action and resistance to colistin are a little superficially described. For example, there is reference to the “membrane of Gram-negative bacteria”. But Gram-negative bacteria have an Outer Membrane and a Cytoplasmic Membrane, and polymyxins interact with both. In the same way, resistance mechanism also could include both membranes. I think this should be improved, in order to improve the scientific rigor (I could suggest this paper, but others are available: DOI 10.1099/mic.0.001136).

3. Line 74: There is an extra comma that should be removed.

RESULTS

4. Results section could be improved if differently organized. For example:

- section 2.1 and the first paragraph of section 2.3 could be joined, as all respects to isolates resistant to colistin, how they were identified as colistin resistant, and the corresponding MICs;

- This could be followed by section 2.4 (the molecular mechanisms behind the colistin resistance phenotypes);

- A third new section could include data regarding the “Co-resistance and corresponding encoding genes to non-polymyxin antibiotics” (that at the moment are presented in the second paragraph of section 2.3);

- Finally, the section 2.2 (genetic relatedness).

5. Line 79: Italicize “K. pneumoniae”.

6. Lines 80-82: This statement is a repeat of the one that apperas at the final of the Introduction section.

7. Line 83: The “and” could be removed.

8. Lines 102-104: This is information that should appear in the Materials and Methods section.

9. Line 109: Title of section 2.2 could be improved to “…K. pneumoniae isolates from human and animal origin”.

10. Line 110: Please replace “Multi locus” by “Multilocus”.

11. Line 112: “belong” or “belonged”? Please use the same verb tenses along the section.

12. Line 116: A reference to Table 1 would be fine here.

13. Line 119: “and” probably should be “any”. Please verify.

14. Line 133: The “antibiotic gradient tests” corresponded to the use of E-test® strips?

15. Line 134: The name of the medium lacks an “e”.

16. Lines 134-135: Any explanation to these different MIC values obtained by different methods?

17. Line 137: This MIC value was obtained by which of the susceptibility methods used?

18. Line 144: The word “respectively” refers to what?

19. Lines 145-147: Some detail regarding which antibiotic resistance genes were detected could be indicated.

20. Line 149: These ESBLs were detected in which isolates? In all? Also, SHV-11 does not provide an ESBL phenotype, so it should be removed.

21. Lines 169-171: Please improve the English.

22. Line 173: Please indicate which those “two mutations” are, also in this paragraph.

DISCUSSION

23. Line 204: Which genes? Only bla genes?

MATERIAL AND METHODS

24. Line 257: “…were included, which were part of a monitoring…”.

25. Line 259-261: The sum totals 272, and not 276. Please check it. 

26. Lines 280: Why blaTEM were not also screened by PCR? Which blaCarbapenemases were searched by PCR?

TABLES

27. Table 1: In the title, it would be better to add “…..K. pneumoniae isolates resistant to COL obtained…..”.

28. Table 2: MIC values corresponding to resistance phenotypes could be highlighted in bold.

29. Supplementary Table S3: In the footnote, please indicate in full what means MHE agar.

Reviewer 2 Report

In this paper the authors identified colistin resistance in veterinary isolates, determined the molecular mechanisms and compared those with isolates obtained from human patients. 

The knowledge of underlying resistance mechanisms of mcr-negative K. pneumoniae colistin resistant veterinary isolates is still incomplete, so their investigation is worthwhile. Strengths of the study include the large number of

isolates and the availability of both susceptibility and molecular data.

The scientific approach seems sound, the results are well presented and overall the paper is well written. I think it will be of interest to the Antibiotics readership.

Author Response

Reviewer 2

Comments and Suggestions for Authors

In this paper the authors identified colistin resistance in veterinary isolates, determined the molecular mechanisms and compared those with isolates obtained from human patients. 

The knowledge of underlying resistance mechanisms of mcr-negative K. pneumoniae colistin resistant veterinary isolates is still incomplete, so their investigation is worthwhile. Strengths of the study include the large number of

isolates and the availability of both susceptibility and molecular data.

The scientific approach seems sound, the results are well presented and overall the paper is well written. I think it will be of interest to the Antibiotics readership.

  • We thank the Reviewer for this positive and motivating feedback!

Reviewer 3 Report

Dear Editor, 

Tietgen et al have presented a detailed report on colistin resistance in K pneumoniae. The authors have identified and compared colistin resistance strains in veterinary and human isolates. The authors also highlight some of the genes in LPS biosynthesis pathway that could be responsible for colistin resistance. 

The authors have presented the data in a clear and concise manner. I do have some questions/suggestions that perhaps the authors could elaborate on. 

1. Have the authors tried MLSA for genetic relatedness. Does the yield the similar results as MLST. 

2. Lines 146-148: Please explain how these were determined 

3. Kindly clarify how A4G point mutation led to K2E mutation. Please provide citation for PROVEAN 

4. Have the authors examined alphafold structures for PmrB of IHIT27665 (T112P) and KP03(P95L), KP54(R256G). It was interesting to see that R256G was the most common mutation in human isolates. 

5. In the discussion it would be useful to include experimental evidence that PmrB mutations cause antibiotic resistance. Recently Kapel N et al published a report of PmrB mutations in P aeruginosa that lead to colistin resistance. 
